# Branched-Chain Amino Acids and Branched-Chain Keto Acids in Hyperammonemic States: Metabolism and as Supplements

**DOI:** 10.3390/metabo10080324

**Published:** 2020-08-09

**Authors:** Milan Holeček

**Affiliations:** Department of Physiology, Charles University, Faculty of Medicine in Hradec Králové, 500 03 Hradec Kralove, Czech Republic; holecek@lfhk.cuni.cz

**Keywords:** glutamine, α-ketoglutarate, urea-cycle disorders, liver cirrhosis, exercise

## Abstract

In hyperammonemic states, such as liver cirrhosis, urea cycle disorders, and strenuous exercise, the catabolism of branched-chain amino acids (BCAAs; leucine, isoleucine, and valine) is activated and BCAA concentrations decrease. In these conditions, BCAAs are recommended to improve mental functions, protein balance, and muscle performance. However, clinical trials have not demonstrated significant benefits of BCAA-containing supplements. It is hypothesized that, under hyperammonemic conditions, enhanced glutamine availability and decreased BCAA levels facilitate the amination of branched-chain keto acids (BCKAs; α-ketoisocaproate, α-keto-β-methylvalerate, and α-ketoisovalerate) to the corresponding BCAAs, and that BCKA supplementation may offer advantages over BCAAs. Studies examining the effects of ketoanalogues of amino acids have provided proof that subjects with hyperammonemia can effectively synthesize BCAAs from BCKAs. Unfortunately, the benefits of BCKA administration have not been clearly confirmed. The shortcoming of most reports is the use of mixtures intended for patients with renal insufficiency, which might be detrimental for patients with liver injury. It is concluded that (i) BCKA administration may decrease ammonia production, attenuate cataplerosis, correct amino acid imbalance, and improve protein balance and (ii) studies specifically investigating the effects of BCKA, without the interference of other ketoanalogues, are needed to complete the information essential for decisions regarding their suitability in hyperammonemic conditions.

## 1. Introduction

Ammonia, derived mainly from the metabolism of amino acids, is toxic when present in high concentrations. Hyperammonemia due to ineffective ammonia detoxification to urea, as occurs in liver injury and urea cycle disorders (UCDs), leads to severe neurological impairments and protein-energy malnutrition development. A marked increase in blood ammonia, resulting from the activated deamination of AMP and amino acid catabolism in the muscles, promotes both central and peripheral fatigue during strenuous exercise [1,2,3]. 

Several studies have demonstrated that high levels of ammonia increase the catabolism of branched-chain amino acids (BCAAs; leucine, isoleucine, and valine) [4,5,6,7], resulting in decreased BCAA levels in liver cirrhosis [8,9,10,11] and UCDs [12,13]. Due to their positive influence on protein balance and the detrimental role of decreased BCAA levels in the pathogenesis of hepatic encephalopathy, BCAAs have been recommended for patients with liver cirrhosis for almost 50 years [8]. Unfortunately, the results of clinical trials do not strongly support the theory that BCAA-containing supplements have beneficial effects [14,15]. Increased ammonia production and BCAA catabolism due to intensive exercise [16,17,18] were the rationale for recommending BCAA-enriched supplements for athletes. Even in these cases, the benefits of BCAA-enriched supplements have not been as great as expected [19,20,21,22,23]. 

It has been suggested that the positive effects of BCAA mixtures are blunted by their increased use in the BCAA aminotransferase reaction to form glutamate, as a pivotal step in ammonia detoxification to glutamine (GLN), in the muscles [24,25,26]. Adverse consequences include the drain of α-ketoglutarate (α-KG) from the tricarboxylic acid (TCA) cycle (cataplerosis) and an increased influx of GLN to the visceral tissues, where it is catabolized into ammonia (Figure 1). Increased ammonia levels after BCAA administration have been reported both in subjects with liver disease [27,28,29,30] and during physical exercise performance [19,20,21,22,23]. 

For many years, in patients with chronic renal failure, ammonia and urea production have been attenuated by the administration of ketoanalogues of essential amino acids (KAEAAs), which can be aminated to the original amino acids [31,32,33]. Hence, the ability of the body to synthesize amino acids from KAEAA creates a theoretical rationale to use branched-chain keto acids (BCKAs)—α-ketoisocaproate (KIC, ketoleucine), α-keto-β-methylvalerate (KMV, ketoisoleucine), and α-ketoisovalerate (KIV, ketovaline)—in order to reduce ammonia production in liver cirrhosis, UCDs, and strenuous exercise. 

The aim of this article is to examine the hypothesis that the adverse effects of BCAA administration on ammonia production and cataplerosis can be attenuated by the administration of BCKAs. For this purpose, I will assess the influence of hyperammonemia on the ability of the body to synthesize BCAAs from BCKAs, along with the results of studies examining the effects of BCKA-enriched supplements in liver cirrhosis, UCDs, and exercise. 

## 2. Amination of BCKAs to BCAAs under Physiological Conditions 

The activity of BCAA aminotransferase, which enables the mutual conversion of BCAAs and BCKAs, is low in the liver and high in muscles. Therefore, most of the exogenous BCAAs (if they are not used for protein synthesis) are converted to the BCKAs in the muscles. The amino group of the BCAA is transferred to α-KG to form glutamate, which then acts as a source of an amino group to form alanine from pyruvate or as a substrate for ammonia detoxification to GLN (Figure 2—left side). Since the activity of BCKA dehydrogenase, which catalyzes the second and irreversible step in the oxidation of BCAAs, is low in the skeletal muscles, most of the BCKAs are released together with GLN and alanine from the muscles into the blood [34]. 

BCKAs released from the muscles are either oxidized or transaminated into BCAAs in several tissues. The second possibility has been clearly demonstrated using radiolabeling in animal experiments [35,36]. Since the BCAA transaminase reaction is reversible and near equilibrium [34], the synthesis of BCAAs from BCKAs may be activated by increased amounts of GLN or alanine released together with the BCKA from the muscles into the blood (Figure 2—right side). This situation occurs during the initial phase of starvation and in various muscle wasting disorders, including sepsis, trauma, surgery, and liver disease. The amination of BCKAs may also be activated by the inhibition of BCKA dehydrogenase, as occurs in the liver during starvation [37]. An interorgan cycle—in which the muscles act as a source of BCKA, alanine, and GLN and other tissues aminate the BCKAs into the corresponding BCAAs—has been postulated [38].

## 3. BCAA Synthesis from BCKAs under Hyperammonemic Conditions 

Due to the equilibrium nature of BCAA aminotransferase, the direction of the net flux of BCAA transamination is determined by the availability of the donors and acceptors of nitrogen [34]. Hence, BCKA amination into BCAAs should be facilitated by a decreased BCAA and increased GLN availability, as occurs in liver cirrhosis, UCDs, and intensive exercise. A study that used isolated perfused livers demonstrated that the decreased delivery of GLN to hepatic tissue decreased the synthesis of BCAAs from BCKAs [39]. A high efficiency of amination of BCKA in subjects with impaired liver function was demonstrated in rat models of severe liver injury and portal-systemic shunting by the increased utilization of labeled KIC for the synthesis of proteins (which indicates previous KIC conversion to leucine) in various organs, including the muscles. In the liver, however, KIC utilization for protein synthesis was unaffected or slightly reduced [40].

## 4. BCAA Synthesis from Exogenous BCKAs

Studies have shown that the exogenous load of BCKAs can shift the BCAA aminotransferase reaction towards BCAA production in several tissues, including the heart, brain, gut, kidneys, liver, and even the skeletal muscle [37,41,42,43]. However, which tissue plays the main role in the amination of exogenous BCKAs is not clear. The route of BCKA administration might have significant influence. 

After oral consumption, enterocytes and the liver, in which GLN is catabolized and glutamate concentrations are nearly three times higher than in the muscles [44], might play a dominant role in the conversion of BCKAs to BCAAs. Abumrad et al. [45] found that 23% of KIC administered into the guts of postabsorptive dogs entered the bloodstream as leucine. The hepatic uptake of KIC was equivalent to 35% of the administered load, and of that, one-third was transaminated into leucine and two-thirds were converted into ketone bodies. Therefore, it may be supposed that most of the BCKAs, which were not converted into BCAAs in the gut and appeared in the portal blood, were oxidized in the liver due to its high BCKA dehydrogenase activity. Nevertheless, Khatra et al. [46] showed that, in patients with alcoholic cirrhosis, there was reduced activity of the hepatic BCKA dehydrogenase to 20% of normal, suggesting that this reduction may enhance the efficiency of oral BCKAs as substitutes for BCAAs. 

The parenteral administration of BCKAs should increase the role of tissues other than the liver and intestine in BCKA amination into BCAAs. The finding in favor of parenteral administration is the higher BCAA levels after BCKA infusion than after oral gavage, as observed by Okita et al. [47] in carbon tetrachloride-treated rats. 

## 5. Benefits of BCKA-Containing Supplements in Hyperammonemic States 

The predicted benefits of BCKA administration in hyperammonemic states include: Decreased ammonia production;The attenuation of cataplerosis;A supply of anaplerotic agents;The correction of amino acid imbalance;Nitrogen sparing and protein anabolism.

### 5.1. Decreased Ammonia Production 

The replacement of BCAAs with BCKAs should decrease ammonia production due to a decreased nitrogen load and by the diversion of glutamate from the glutamate dehydrogenase reaction, in which ammonia is liberated, to synthesize BCAAs from BCKAs (see Figure 3). 

### 5.2. Attenuation of Cataplerosis 

The formation of α-KG from glutamate during the amination of BCKAs attenuates the drain of α-KG from the TCA cycle (cataplerosis), activated by the detoxification of ammonia into GLN in the muscles (see Figure 1 and Figure 3). 

### 5.3. Supply of Anaplerotic Agents 

BCKAs, which are not aminated to a corresponding BCAA, are catabolized. The catabolism of KIC leads to acetyl-CoA and acetoacetate formation (KIC is ketogenic); KIV is catabolized into succinyl-CoA (KIV is glucogenic), and KMV, to acetyl-CoA and succinyl-CoA (KMV is both glycogenic and ketogenic). Therefore, intermediates of KIV and KMV catabolism can enter the TCA cycle and act as anaplerotic substances, but not KIC metabolites (see Figure 3).

### 5.4. Correction of Amino Acid Imbalance

Subjects with liver cirrhosis usually have a decreased ratio between BCAA concentrations and aromatic amino acid (AAA; tyrosine, phenylalanine, and tryptophan) concentrations, which plays a role in the pathogenesis of hepatic encephalopathy [8]. The amination of BCKAs into BCAAs can correct the imbalance between the BCAAs and AAAs and attenuate signs of encephalopathy.

Unfortunately, since the capacity of the body for BCKA decarboxylation is higher than that for amination, a high portion of exogenous BCKAs are catabolized. Schauder et al. [48] postulated that if BCKA-containing supplements were given to increase low BCAA concentrations, the dose would need to be significantly higher than that for BCAA-containing supplements. Therefore, in most conditions, BCKAs should not be considered as substances for separate use but rather as substances that suitably replace a part of the BCAA. 

### 5.5. Nitrogen Sparing and Protein Anabolism 

Significant numbers of data demonstrate the nitrogen-sparing effects of BCKAs. The BCKAs improved protein balance in isolated rat muscles, parenterally fed rats, fasting obese men, and patients undergoing major abdominal surgery and with Duchenne muscular dystrophy [36,49,50,51,52]. The effects can be mediated by the BCAAs—particularly leucine, which stimulates protein synthesis through the PI3K/Akt/mTOR signaling pathway [50]—and by ketone bodies generated in the metabolism of KIC and KMV [53]. A role may also be played by β-hydroxy-β-methylbutyrate (HMB), which is synthesized from KIC in the liver, decreases the activity of the ubiquitin-proteasome proteolytic pathway, and exerts protein anabolic effects in the muscles [54]. 

AAA, aromatic amino acids (tyrosine, phenylalanine, and tryptophan); HMB, β-hydroxy-β-methylbutyrate.

## 6. Effects of BCKAs in Subjects with Liver Disease

Although there is a good theoretical basis for recommending BCKAs for the treatment of patients with liver disease, the number of reports is very small and most are dated to the last century (Table 1). Nevertheless, these studies have provided proof that subjects with liver disease can effectively synthesize amino acids from their ketoanalogues [9,47,55]. Some studies have reported beneficial effects on the signs of hepatic encephalopathy [9,55]. Unfortunately, there are no studies reporting effects of BCKAs on nutritional status and examining in which stage of liver disease their administration might be optimal.

## 7. Effects of KAEAA on UCDs

UCDs are a group of inborn errors of metabolism associated with a defect in or deficiency of any of the enzymes involved in urea synthesis. These disorders are characterized by profound hyperammonemia, seizures, lethargy, coma, and death in the neonatal period or severe long-term neurological impairment. A marked decrease in BCAAs is observed in acute metabolic decompensation [13] and patients treated with phenylbutyrate, a drug used to decrease ammonia, which, unfortunately, activates BCKA dehydrogenase, resulting in increased BCAA oxidation in the liver and muscles [59,60,61].

Early approaches to therapy for UCDs with KAEAAs in the 1970s clearly demonstrated their effective amination and potential to manage hyperammonemia [12,62,63,64,65,66]. Decreased levels of ammonia, improved clinical status, and nitrogen-sparing effects of KAEAAs have been observed in carbamoyl phosphate synthetase deficiency, citrulinemia, and ornithine transcarbamylase deficiency (Table 2). Unfortunately, despite the short-term success of the therapy, most of the infants died within the first year of life from acute hyperammonemic episodes or other complications of the disease. The limitations of all these studies is the use of mixtures adopted for subjects with uremia, and studies specifically examining the effects of BCKAs are not available. 

KAEAAs are not currently used in the treatment of hyperammonemia in UCDs. The principal therapeutic regimens include nitrogen-scavenger drugs (benzoate or phenylbutyrate), l-arginine, and hemodialysis.

## 8. Effects of BCKAs and KAEAAs in Exercise 

Exercise greatly increases energy expenditure, promotes the oxidation of BCAAs, and increases ammonia and GLN levels in the blood due to the activated deamination of AMP to inosine-5-monophosphate and amino acid catabolism in muscles [16,17,18,67,68,69]. There is consensus that exercise-induced hyperammonemia contributes to both central and peripheral fatigue [1,2,3]. Studies with 13C-labeled leucine showed that the oxidation of BCAAs increased 2- to 3-fold during exercise [16,17]. 

BCAAs are recognized as supplements for athletes with a number of benefits, notably on muscle protein synthesis, fatigue, and exercise-induced muscle damage [70,71]. However, unfortunately, several studies have failed to demonstrate the benefits of BCAA supplementation [72,73,74]. In addition, BCAA administration increases blood ammonia levels during and after exercise, suggesting undesirable effects on muscle performance [19,20,21,22,23]. 

Studies performed in more recent years have shown that BCKA and KAEAA supplementation improves muscle performance and attenuates exercise-induced hyperammonemia and muscle damage (Table 3). Unfortunately, studies specifically examining the effects of BCKAs are rare. 

## 9. Conclusions

The studies examining the metabolism and physiological role of the BCKAs, along with reports of the positive effects of BCKAs, KAEAAs, and AA + KAEAAs in subjects in various hyperammonemic states, indicate the overlooked therapeutic potential of BCKAs. The administered BCKAs should remove excess body nitrogen by being aminated to corresponding BCAAs and, in this way, attenuate ammonia production and increase the BCAA/AAA ratio, which plays a role in the pathogenesis of hepatic encephalopathy. BCKAs may act also as anaplerotic agents, ameliorate the impairment of the TCA cycle and ATP production, and improve the protein balance of the body.

The shortcoming of many of the reports explored in this article is that the BCKAs were not administrated alone but instead in association with other KAEAAs or with amino acids (AA+KAEAAs), particularly in the form of mixtures developed for the therapy of renal insufficiency, which are unsuitable for patients with liver disease. This is particularly true for the contents of tryptophan and 2-oxo-phenylpropionoc acid (precursor of phenylalanine and tyrosine), which can worsen the signs of hepatic encephalopathy [8]. 

Potential areas for further research include searching for an optimal ratio among individual BCKAs, as well as determining which form of BCKA is optimal. When searching for optimal ratios, the differences in the metabolism of individual BCKAs might be important. KIV and KMV can be used as anaplerotic agents, which can attenuate the cataplerosis of α-KG, but not KIC, which is exclusively ketogenic. The forms of BCKAs examined in clinical studies to date include sodium, calcium, and ornithine salts. Unfortunately, studies attempting to compare individual substances are few. Of interest should be the study of Herlong et al. [9], which demonstrated that ornithine salts of the BCKAs improved chronic portal-systemic encephalopathy more than their components given separately and more than BCAAs.

In conclusion, therapeutic strategies are needed to ameliorate the adverse effects of BCAAs on cataplerosis and ammonia formation. Reports on the effects of BCKAs on ammonia and BCAA levels, and various parameters of the nutritional state in liver disease, UCDs, and exercise are encouraging and indicate that BCKAs may offer therapeutic advantages over their respective BCAAs. Further studies specifically examining the effects of BCKAs without interference from other KAEAAs, which may blunt benefits of the BCKAs, are needed to complete the information essential for the proper administration of these compounds in hyperammonemic conditions.

## Figures and Tables

**Figure 1 metabolites-10-00324-f001:**
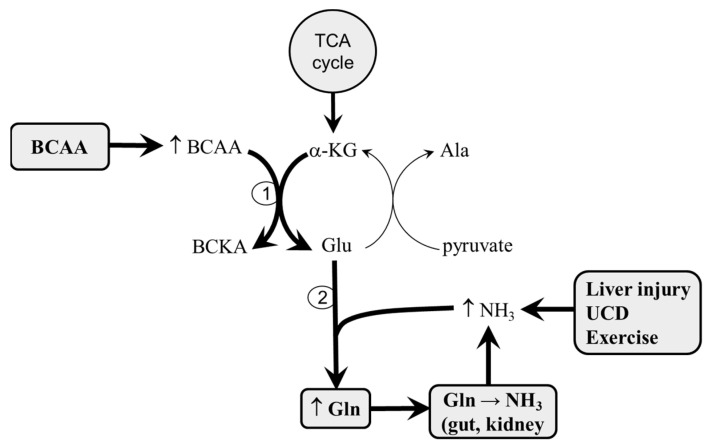
Supposed effects of branched-chain amino acid (BCAA) administration in hyperammonemic conditions. In hyperammonemic conditions, most of the exogenous BCAAs are used for the synthesis of glutamate, which is a direct substrate for ammonia detoxification to glutamine (GLN). The results are the diversion of α-ketoglutarate (α-KG) from the TCA cycle (cataplerosis) and GLN catabolism to ammonia in the visceral tissues (particularly the gut and kidneys). When the detoxification of ammonia to urea is compromised, a vicious cycle, in which enhanced ammonia concentrations activate GLN synthesis, is activated [25].

**Figure 2 metabolites-10-00324-f002:**
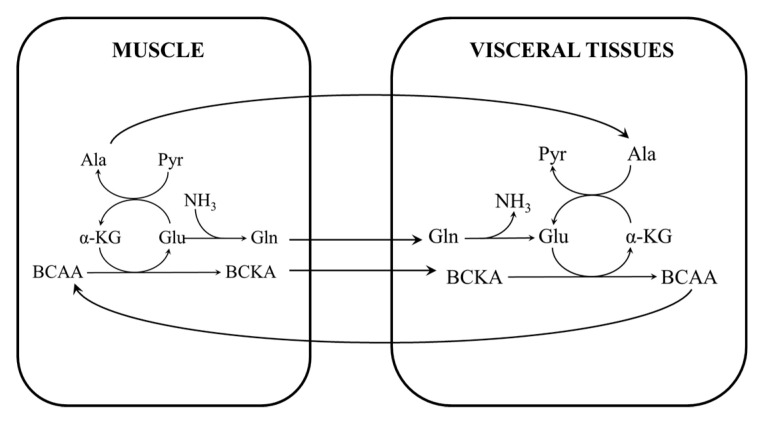
A scheme of the BCAA–branched-chain keto acid (BCKA) cycle between the muscles and visceral tissues. In the muscles, BCAAs are the main donor of nitrogen to α-KG to form glutamate, which may be converted to alanine or act as a substrate for GLN synthesis. The main sources of nitrogen for the amination of BCKAs to BCAAs in the visceral tissues are GLN, glutamate, and alanine.

**Figure 3 metabolites-10-00324-f003:**
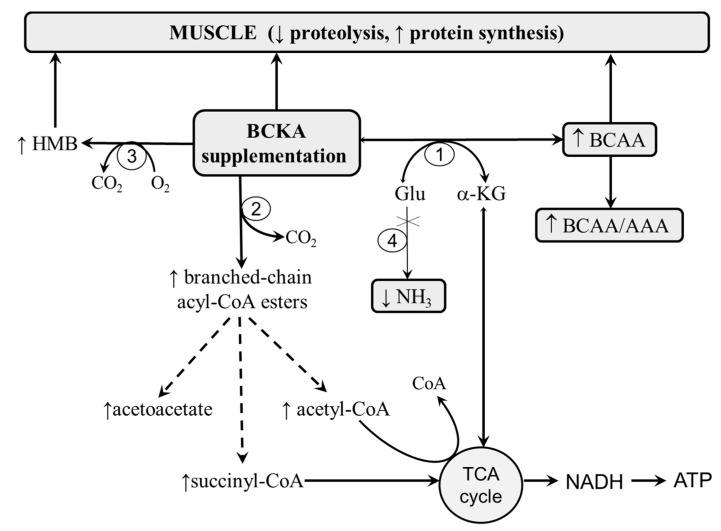
Supposed effects of BCKA supplementation on (i) the formation of BCAAs, (ii) diversion of glutamate from breakdown to ammonia to the BCAA and α-KG synthesis, (iii) protein balance, and (iv) the TCA cycle (anaplerosis). 1, branched-chain aminotransferase; 2, BCKA dehydrogenase; 3, ketoisocaproate (KIC) dioxygenase; 4, glutamate dehydrogenase.

**Table 1 metabolites-10-00324-t001:** Effects of BCKAs and ketoanalogues of essential amino acids (KAEAAs) in subjects with liver disease.

Study Design	Results	Reference
Patients with cirrhosis and PSE; AA + KAEAA, infusion (1–5 days) or orally (3–12 days).	↑ BCAA, methionine and phenylalanine, ↓ glutamine. No effect on ammonia levels. Improvement in mental status and psychological testing.	[55]
Patients with cirrhosis and PSE; BCAA, ornithine, or calcium salts of BCKA orally for 7–10 days; double-blind crossover comparison.	Combination of ornithine and BCKA improved EEG and clinical signs of PSE more than BCAA or components given separately.	[9]
Patients with cirrhosis and healthy controls; KIC infusion (300 µmol/min for 150 min).	↑ leucine and ammonia; ↓ urea, valine, isoleucine, methionine, phenylalanine, and GLN.	[56]
Patients with cirrhosis and PSE; lactulose and protein restriction; BCKA (15 g/day) or placebo for 4 weeks in a crossover regimen.	No effect on ammonia and BCAA levels, EEG, number connection test, and clinical state.	[57] *
Rats, acute liver injury (CCl_4_); BCKA (sodium or ornithine salts); infusion (60 min) or intragastric administration.	Higher BCAA levels after BCKA in CCl_4_-treated animals than in controls after infusion. Only slight increases in BCAAs after gavage of BCKAs.	[47]

AA + KAEAA, mixture of amino acids and ketoanalogues of essential amino acids; PSE, portal systemic encephalopathy. * The possible cause of the unobserved rise in the BCAAs after BCKA administration is that the collection of blood samples was performed in a post-absorptive state, in which most of the changes induced by BCAA supplementation disappear [58].

**Table 2 metabolites-10-00324-t002:** Effects of KAEAAs in urea cycle disorders (UCDs).

Study Design	Results	Reference
Carbamoyl phosphate synthetase deficiency; KAEAAs administered by infusion or orally.	Infusion: ↓ ammonia and ↑ BCAAs, methionine, and phenylalanine. Oral intake: ↓ ammonia and alanine.	[62]
Carbamoyl phosphate synthetase deficiency; AA + KAEAAs orally for one year.	↓ ammonia and ↑ BCAAs, methionine, and phenylalanine; improved clinical status. ↑ ammonia, GLN, and alanine after withdrawal.	[12]
Citrullinemia; EAA + KAEAAs orally for 8 months.	↓ ammonia and citrulline until death due to diarrhea and dehydration.	[63]
Citrullinemia; AA + KAEAAs orally for 7 months.	↓ ammonia until death due to acute hyperammonemic crisis.	[64]
Ornithine transcarbamylase deficiency; compared effects of LPD + KAEAAs, LPD + EAAs, and LPD + lactulose.	LPD + KAEAAs better than LPD + EAAs or LPD + lactulose.	[65]
Ornithine transcarbamylase deficiency; KAEAAs orally.	Ammonia, growth, and development maintained near normal from 2nd day until death at 5 months.	[66]

KAEAAs, ketoanalogues of essential amino acids; AA + KAEAAs, mixture of amino acids and KAEAAs; EAA + KAEAAs, mixture of essential amino acids and KAEAAs; LPD, low protein diet.

**Table 3 metabolites-10-00324-t003:** Effects of KAEAAs and BCKAs on ammonia in exercise.

Study Design	Results	Reference
Male patients with McArdle’s disease; BCAAs or BCKAs prior to start cycling exercise.	After BCAAs: deterioration of exercise performance and ↑ in ammonia. After BCKAs: improved exercise performance and smaller ↑ in ammonia.	[24]
Rats; AA+KAEAAs (0.3 g/kg) or saline orally 1 h before exercise.	Attenuated increase in ammonia; ↓ urea.	[75]
Cyclists; ketogenic diet for 2 days before experiment, AA+KAEAAs or lactose orally 1 h before cycling (2 h).	Attenuated increase in ammonia induced by exercise.	[76]
Male untrained volunteers; α-KG or BCKAs (0.2 g/kg/d) for 4 weeks during endurance training (running).	α-KG or BCKAs improved training effects and recovery state.	[77]
Patients with type 2 diabetes; training on cycle ergometer and mixture of α-KG and BCKAs (0.2 g/kg orally) for 6 weeks or placebo (glucose, sodium and calcium salts).	Positive effects on physical training (higher VO_2max_, endurance capacity, and power output).	[78]
Cyclists; ketogenic diet for 2 days before experiment, AA+KAEAAs or lactose, cycling session (2 h) followed by a maximum test.	↑ (~70%) ammonia in placebo, not in AA+KAEAAs group. No difference in physical or cognitive performance.	[79]
Cyclists, ketogenic diet for 2 days before experiment, AA+KAEAAs or lactose orally 1 h before cycling (2 h).	↑ ammonia, creatine kinase, lactate dehydrogenase, and AST in placebo group. No significant changes in AA+KAAAs group.	[80]

AA + KAEAAs, mixture of amino acids and ketoanalogues of essential amino acids.

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
