# Peer review of "Branched-Chain Amino Acids and Branched-Chain Keto Acids in Hyperammonemic States: Metabolism and as Supplements"

_metabolites, 2020, doi:10.3390/metabo10080324_

Round 1

Reviewer 1 Report

„Chapter 5 Benefits of BCKA-containing supplements in hyperammonemic states”

Could author add any references and study examples where BCKA -supplements were used. Especially add data concerning the changes in changes in protein level/ status in patients.

Additionally in which stage of liver disfunction  for patients is better introduce the BCKA supplements. Authors generally have written in hyperammonemia. But before hyperammonemia patients may have pure nutritional status and high protein deficiency.

Author Response

I would like to thank the reviewer for constructive and helpful comments. My response is outlined below.

Comment

„Chapter 5 Benefits of BCKA-containing supplements in hyperammonemic states”

Could author add any references and study examples where BCKA -supplements were used. Especially add data concerning the changes in changes in protein level/ status in patients.

Response

The following sentence has been added into the revised version of the manuscript: ”The BCKAs improved protein balance in isolated rat muscles, parenterally fed rats, fasting obese men, and patients undergoing major abdominal surgery and with Duchenne muscular dystrophy [36,49-52]. See page 5, lines 172-174.

Comment

Additionally in which stage of liver disfunction for patients is better introduce the BCKA supplements. Authors generally have written in hyperammonemia. But before hyperammonemia patients may have pure nutritional status and high protein deficiency.

Response

Thanks for suggestion. In the revised version is stated: ”Unfortunately, there are no studies reporting effects of BCKAs on nutritional status and examining in which stage of liver disease might be optimal”. See page 5, lines 195-196.

Reviewer 2 Report

Review of manuscript ID: metabolites-894893

Title: Branched-chain amino acids and branched-chain keto acids in hyperammonemic states: Metabolism and as supplements.

Summary:

This is a well written review by M. Holeček. The review discusses the ability of the body to synthesize BCAAs from BCKAs and highlights studies assessing effects of BCKA supplementation in hyperammonemic states including: liver cirrhosis, UCDs, and exercise.

Minor comments to the author:

  1. Page 3, line 111: the sentence “Significant influence might have the route of BCKA administration.” Needs to be reworded. Perhaps to: the route of BCKA administration may have significant influence.
  2. Page 6, line 194: I would change the word “shortage” to limitations “of all these studies is the use of mixtures …”
  3. Table 2, results column: for reference 64 do you mean continued failure to thrive?
  4. Page 7, line 205: this sentence needs to be re-worded. It is confusing. “The main role in pathogenesis of increased ammonia levels in the blood has probably the activated deamination of AMP to inosine-5-monophosphate and amino acid catabolism in muscles”
  5. Page 7, line 214: I suggest changing the word reports to studies “performed in more recent years …”
  6. Page 7, line 223: remove the word to from the sentence: “in this way to attenuate ammonia production”
  7. Page 8, line 248: Are these sentences an error? “This section may be divided by subheadings. It should provide a concise and precise description of the experimental results, their interpretation as well as the experimental conclusions that can be drawn.”

Author Response

Response

I would like to thank the reviewer for constructive and helpful comments. My response is outlined below.

Comment

  1. Page 3, line 111: the sentence “Significant influence might have the route of BCKA administration.” Needs to be reworded. Perhaps to: the route of BCKA administration may have significant influence.

Response

Done as suggested. P. 3, l. 117. Thank you.

Comment

  1. Page 6, line 194: I would change the word “shortage” to limitations “of all these studies is the use of mixtures …”

Response

Done as suggested. Thank you. Please see p. 6, line 226.

Comment

  1. Table 2, results column: for reference 64 do you mean continued failure to thrive?

Response

Thanks. The sentence has been modified as follows: “↓ ammonia until death due to acute hyperammonemic crisis.” See Table 2, page 7.

Comment

  1. Page 7, line 205: this sentence needs to be re-worded. It is confusing. “The main role in pathogenesis of increased ammonia levels in the blood has probably the activated deamination of AMP to inosine-5-monophosphate and amino acid catabolism in muscles”

Response.

This redundant sentence has been removed and the paragraph amended to make it easier to understand. Page 7, lines 239-242.

Comment

  1. Page 7, line 214: I suggest changing the word reports to studies “performed in more recent years …”

Response

Done as suggested. P. 7, line 248.

Comment

  1. Page 7, line 223: remove the word to from the sentence: “in this way to attenuate ammonia production”

Response

Done as suggested. P. 7, line 269.

Comment

  1. Page 8, line 248: Are these sentences an error? “This section may be divided by subheadings. It should provide a concise and precise description of the experimental results, their interpretation as well as the experimental conclusions that can be drawn.”

Response

The sentence was the text of the template and was deleted.